# The genomic ecosystem of transposable elements in maize

**Michelle C. Stitzer** [1]\*, **Sarah N. Anderson** [2], **Nathan M. Springer** [2], **Jeffrey Ross-Ibarra** [1,3]

**1** Center for Population Biology and Department of Evolution and Ecology, University of California, Davis, California, United States of America, **2** Department of Plant and Microbial Biology, University of Minnesota, Saint Paul, Minnesota, United States of America, **3** Genome Center, University of California, Davis, California, United States of America

\* mcstitzer@ucdavis.edu

**Data Availability Statement:** All relevant data are within the manuscript and its Supporting information files. Scripts for generating summaries from data sources and links to summarized data

## Abstract

Transposable elements (TEs) constitute the majority of flowering plant DNA, reflecting their tremendous success in subverting, avoiding, and surviving the defenses of their host genomes to ensure their selfish replication. More than 85% of the sequence of the maize genome can be ascribed to past transposition, providing a major contribution to the structure of the genome. Evidence from individual loci has informed our understanding of how transposition has shaped the genome, and a number of individual TE insertions have been causally linked to dramatic phenotypic changes. Genome-wide analyses in maize and other taxa have frequently represented TEs as a relatively homogeneous class of fragmentary relics of past transposition, obscuring their evolutionary history and interaction with their host genome. Using an updated annotation of structurally intact TEs in the maize reference genome, we investigate the family-level dynamics of TEs in maize. Integrating a variety of data, from descriptors of individual TEs like coding capacity, expression, and methylation, as well as similar features of the sequence they inserted into, we model the relationship between attributes of the genomic environment and the survival of TE copies and families. In contrast to the wholesale relegation of all TEs to a single category of junk DNA, these differences reveal a diversity of survival strategies of TE families. Together these generate a rich ecology of the genome, with each TE family representing the evolution of a distinct ecological niche. We conclude that while the impact of transposition is highly family- and context-dependent, a family-level understanding of the ecology of TEs in the genome can refine our ability to predict the role of TEs in generating genetic and phenotypic diversity.

## Author summary

Transposable elements (TEs) are pieces of DNA that can jump to new positions in the genome. When they land at a new location, they generate a mutation. Such mutations in genes affecting kernel and plant pigmentation allowed the discovery of TEs in maize in the 1940's. Since, we have learned that TEs are a ubiquitous feature of eukaryotic genomes, and that TEs make up over 85% of all the DNA in a maize genome. Here, we investigate

are available at http://www.github.com/mcstitzer/maize_genomic_ecosystem.

**Funding:** M.C.S. and J.R.-I. are supported by the National Science Foundation Plant Genome award 1238014. M.C.S. acknowledges support from the National Science Foundation Graduate Research Fellowship under Grant No. 1650042; J.R.-I. acknowledges support from the USDA Hatch project CA-D-PLS-2066-H. S.N.A. and N.M.S. are supported by a grant from USDA-NIFA (2016-67013-24747). The funders had no role in study design, data collection and analysis, decision to publish, or preparation of the manuscript.

**Competing interests:** The authors have declared that no competing interests exist.

the roles of individual TE copies and TE species interacting within the maize genome, and how these relationships are analogous to ecological communities. The community of transposable elements within a single genome represents a rich diversity of ecological strategies for survival in the complex, hostile environment of a host genome.

## Introduction

*'Lumping our beautiful collection of transposons into a single category is a crime'*

-Michael R. Freeling, Mar. 10, 2017

Transposable elements (TEs) are pieces of DNA that can replicate or move themselves within a genome. The majority of DNA in plant genomes is TE derived, and their activity is the largest contributor to differences in genome size within and between taxa [1]. When they transpose, TEs also generate mutations as they insert into novel positions in the genome [2, 3]. These two linked processes—that of replication of the TE, and mutation suffered by the host genome—generate a conflict between individual lineages of TEs and their host genome. Individual TE lineages gain evolutionary advantage by increasing in copy number, while the host genome gains fitness if it can reduce deleterious mutations arising from transposition. As a result of this conflict, many genomes are littered with a bulk of TE-derived DNA that is often both transcriptionally and recombinationally inert [4]. While this conflict between TEs and their host has long been noted to shape general patterns of TE evolution [5–8], the details of how this conflict unfolds are tenuous and rarely well understood [9, 10].

The staggering diversity of TEs presents a major challenge in understanding their conflict with the host genome. For example, although they are united by their ability to move between positions in the host genome, the mechanisms by which TEs do so differ between the major TE classes. Class I retrotransposons, often the major contributor of TE DNA in plants [11], can be further divided into three orders—long terminal repeat (LTR), long interspersed nuclear element (LINE), and short interspersed nuclear element (SINE). All class I TEs are transcribed to mRNA by host polymerases, some are translated to produce reverse transcriptase and other enzymes, and all use TE encoded enzymes for reverse transcription of a cDNA copy that can be integrated at a new position in the host genome. In contrast, the two major orders of class II DNA TEs transpose in different ways. TIR elements are physically excised from one position on the chromosome and moved by TE-encoded transposase proteins that recognize short, diagnostic, terminal inverted repeats (TIRs). Helitron elements are thought to transpose via a rolling circle mechanism that generates a new copy after a single strand nick by an element-encoded protein and subsequent strand invasion and repair [12]. The process of transposition for most TEs (all LTR, TIR; some LINE, SINE) generates a target site duplication (TSD) in the host DNA at the integration site, and thus the identification of a TSD bordering a TE can confirm transposition. These well-described mechanisms of transposition generate predictable sequence organization that can be recognized computationally, but also generate differences in the genomic localization of these elements, via enzymatic site preference of TE encoded proteins [13, 14]. Most of these orders are further subdivided into TE superfamilies based on differences in the sequence, arrangement, and function of proteins encoded by the TE to ensure its transposition [15].

The process of transposition generates new TE copies within a genome, forming relationships between TEs that allow their systematic grouping into families. Many taxonomic

schemes for TEs exist [15–19], but the most widely-applied approach for genome-scale data [15] relies on sequence homology between copies. Although not entirely representative of TE evolutionary history [20, 21], such approaches nonetheless reflect to some degree the ability of TE encoded proteins to bind TE DNA and move other TE copies in *trans*, as recognition of specific nucleic acid sequences by TE-encoded proteins is a necessary step in the transposition process. The resulting TE families thus represent groups of related TEs that share both evolutionary history and transposition machinery, and are the groupings most naturally analogous to species in higher eukaryotes.

TE families differ from one another in many ways, including their total copy number, where they insert in the genome, which tissues they are expressed in, and how they are restricted epigenetically by the host genome. In the maize genome, some families are small, consisting of a few (e.g. *Bs* [22]) or tens (e.g. *Ds1* [23]) of copies, while others contain tens of thousands of copies (e.g. *huck*, *cinful-zeon* [24–28]). Some TE families are expressed in certain tissues (e.g. *Misfit* [29]), while others are expressed more broadly across many tissues (e.g. *cinful* [29]). Some families preferentially insert into genic regions (e.g. *Mu1* [30]), others in the centromere (e.g. *CRM1* [31]). And while some families lack DNA methylation, others are methylated across the entire body of the TE, and yet others act to spread methylation outwards into flanking sequences [32, 33].

Although the major classes of TEs are found across taxa, their relative abundances differ [34] and there is no clear consensus as to the factors that explain the diversity of TEs within a genome [35–38]. One approach to understand the diversity of TEs is to consider the genome as a community and apply principles of community ecology to understand their distribution and abundance [39]. Initially proposed in terms of a dichotomy between TEs that have specialized in heterochromatic or euchromatic niches [7], thoughts about the ecology of the genome have been refined into a continuum of space, with different TE lineages existing in different genomic niches [8, 39, 40]. Empirical descriptions of TEs in a community ecology context, however, have been limited to a few families [41–43]. The genome reflects the interface between ecological and evolutionary processes, as TEs alter their environment by inserting. This in turn affects how TEs evolve and adapt—making the genome a system to explore the interface between ecology and evolution [44–46]. Due to the long time scales over which TEs can persist in genomes, distinguishing whether processes occurring within the genome reflect ecological or evolutionary time scales can be difficult, although the two can be separated [47].

Here, we utilize the genomic ecosystem as a framework to describe patterns we observe in the extant maize genome. We take advantage of the diversity of TEs in maize, the record of past transposition still detectable in the genome, and the rich developmental and tissue-specific resources of maize to investigate the family-level ecological and evolutionary dynamics of TEs in maize. We integrate many metrics that can be measured at the level of TE family to present a natural history of TEs in the B73 maize genome, characterizing and describing the genomic features that differentiate superfamilies and families of TEs. We model survival of individual copies and families in the genome to facilitate an understanding of the complex and interactive strategies TEs use to associate with their host and each other, and identify suites of traits that act to define specific genomic niches and survival strategies. We conclude that understanding the diversity of TEs in the maize genome helps not only to describe TE function, but also that of the host genome.

## Results

### General features of TE orders and superfamilies

We identified members of each of the 13 superfamilies of transposable elements (TEs) previously identified in plants [15] in our structural annotation of the maize B73 reference genome.

This annotation resolves nested insertions of TEs within other elements, resulting in a total of 143,067 LTR retrotransposons (RLC (*Ty1/Copia*), RLG (*Ty3/Gypsy*), and RLX (*Unknown LTR*) superfamilies), 1,640 LINE and SINE (nonLTR) retrotransposons (RIL, RIT, and RST superfamilies), 171,570 TIR transposons (DTA (*hAT*), DTC (*CACTA*), DTH (*Pif/Harbinger*), DTM (*Mutator*), DTT (*Tc1/Mariner*), and DTX (*Unknown TIR*) superfamilies), and 22,234 Helitrons (DHH superfamily) (Table 1 and Fig 1A). We determined the number of families, median length, median age, distance to the nearest gene, and the number of base pairs each superfamily contributes to the genome (Fig 2; Interactive distributions per family: https://mcstitzer.shinyapps.io/maize_te_families/). For each family and superfamily, we determined the proportion of elements that are nested within another TE and the proportion of elements that are split into multiple pieces by other TE insertions.

Even at the broad taxonomic level of order, there are considerable differences among TEs. Because of their size, (median length 8.4 kb; Fig 2A and S1(C) Fig) LTR retrotransposons contribute more total base pairs to the genome (1,363 Mb; Fig 1B) and are commonly disrupted by another TE copy ($\approx \frac{2}{3}$ disrupted; S1(D) Fig). LTR retrotransposons are also typically far from genes (median distance 16.4 kb; Fig 2B, only 3.5% within a gene transcript; S1(A) Fig, median distance to a syntenic gene 31.9 kb; S1(B) Fig) and $\approx \frac{1}{2}$ of copies insert into a preexisting TE copy (Fig 2C). The median time since insertion of LTR retrotransposons is 315,000 years (Fig 2D). In contrast, despite having more copies (Table 1), TIR elements contribute fewer base pairs to the genome (74.1 Mb) and are rarely disrupted by the insertion of another TE copy ($< 5\%$ disrupted) (S1(D) Fig), presumably due to their much smaller size (median length 306 bp; Fig 2A and S1(C) Fig). TIR elements as a group are slightly further from genes (median distance 17.2 kb; Fig 2B, 1.7% within a gene transcript; S1(A) Fig, median distance to a syntenic gene 29.0 kb, S1(B) Fig), and commonly insert into preexisting TE copies ($\approx 70\%$ of copies; Fig 2C). They represent the most recent insertions, with a median age of 185,000 years (Fig 2D). Although Helitron elements are fewer in number than TIR elements, they contribute more base pairs to the genome (93.8 Mb) and are more commonly disrupted by the insertion of another TE ($\approx \frac{1}{4}$ of copies; S1(D) Fig) due to their increased length (median length 2.4 kb). Helitrons are also closer to genes than TIR elements (median distance 10.4 kb; Fig 2B, with 22.9% overlapping a gene transcript S1(A) Fig; median distance 25.4 kb from a syntenic gene; S1(B) Fig), and less frequently insert into a preexisting copy (50% of copies are found within another TE; Fig 2C). Helitrons are represented by relatively old copies, with a median age of 500,000 years (Fig 2D). NonLTR retrotransposons (LINEs and SINEs) contribute only 2.9 Mb, are relatively short (median length 548 bp), and only 5% of copies are disrupted by the insertion of another TE (S1(D) Fig). LINEs and SINEs are however often close to genes (median distance 2.3 kb; Fig 2B, 18.6% in a gene transcript; S1(A) Fig, median distance to a syntenic gene 10.1 kb; S1(B) Fig) and only 37% insert into another TE copy (Fig 2C). These nonLTR retroelements have a median age of 350,000 years (Fig 2D).

Within these orders, variation also exists among superfamilies (Fig 2). For example, TE superfamilies are found nonuniformly along chromosomes (Fig 3 and S2 Fig): while some superfamilies like RLG (Ty3/Gypsy) and DTC (CACTA) are enriched in centromeric and pericentromeric regions, others, like RLC (Ty1/Copia) and DTA (hAT) are found more commonly on chromosome arms. As maize genes are enriched on chromosome arms, this distribution is reflected in the distance each superfamily is found from genes (Fig 2B). Similarly, while most TIR superfamilies are found far from genes (median 17.2 kb), DTM (Mutator) elements are only a median distance of 2.4 kb away from genes (Fig 2B). And although TIR elements are often short (median 311 bp), DTC elements have a median length of 2886 base pairs (Fig 2A).

**Table 1. TE superfamilies in the maize genome.**

| Class | Order | Superfamily | Common Name | Number Copies | Number Families |
|---|---|---|---|---|---|
| DNA transposon | Helitron | DHH | Helitron | 22,339 | 1,722 |
| DNA transposon | TIR | DTA | hAT | 5,096 | 275 |
| DNA transposon | TIR | DTC | CACTA | 2,768 | 73 |
| DNA transposon | TIR | DTH | Pif/Harbinger | 63,216 | 458 |
| DNA transposon | TIR | DTM | Mutator | 928 | 67 |
| DNA transposon | TIR | DTT | Tc1/Mariner | 67,533 | 269 |
| DNA transposon | TIR | DTX | Unknown TIR | 34,778 | 76 |
| Retrotransposon | LTR | RLC | Ty1/Copia | 46,553 | 2,788 |
| Retrotransposon | LTR | RLG | Ty3/Gypsy | 75,761 | 7,719 |
| Retrotransposon | LTR | RLX | Unknown LTR | 20,789 | 13,290 |
| Retrotransposon | nonLTR | RIT | RTE | 296 | 2 |
| Retrotransposon | nonLTR | RIL | L1 | 477 | 29 |
| Retrotransposon | nonLTR | RST | SINE | 892 | 533 |

## Features of TE families

These descriptive statistics measured at the order and superfamily level are an aggregate across many TE families. TE families are defined based on sequence homology between copies [48], using a 80% sequence similarity cutoff described in Wicker et al. (2007) [15]. This results in thousands of families of LTR retrotransposon and Helitron elements, and hundreds of families of DNA TIR elements (Table 1). Although the majority of all TE families have fewer than ten copies (Fig 1A), the largest LTR retrotransposon and Helitron families in the genome consist of thousands of copies. Consistent with previous analyses built on subsets of maize bacterial

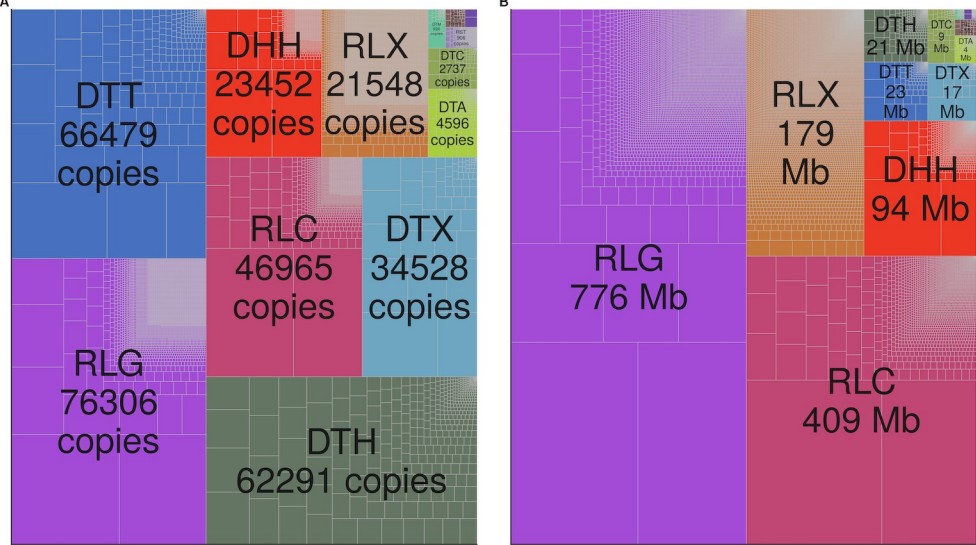

**Fig 1. Abundance of TEs.** The relative copy number (A) and size in million base pairs (Mb) (B) of families and superfamilies shown by the size of the rectangle. Superfamilies are denoted by color, and each family is bounded by gray lines within the superfamily. Superfamily names begin with a two letter code: 'DT' belong to the order of Terminal Inverted Repeat transposons, 'DH' refers to the order Helitron, 'RL' belong to the order Long Terminal Repeat retrotransposons, and 'RI' and 'RS' are nonLTR retrotransposons (LINEs and SINEs). Superfamily names beginning with 'D' are Class II DNA transposons, while those starting with 'R' are Class I retrotransposons.

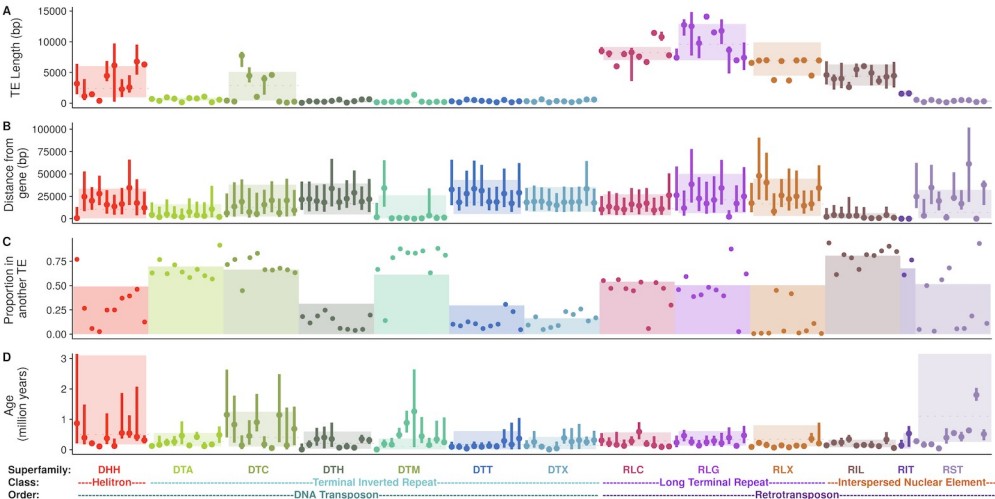

**Fig 2. Characteristics of each superfamily of TE.** Superfamilies are classified into orders and classes, as shown at the bottom of the plot. (A-D) Family characteristics of each of the most numerous 10 families (with $\geq$ 10 copies) of each superfamily. Family names are listed in S1 Table. (A) TE length, (B) Distance to the closest gene, (C) Family proportion of TE copies found within another TE, and (D) TE age. In (A, B, & D) family medians are shown as points, with lines representing upper to lower quartiles. Superfamilies are shown as colored rectangles, where the dotted line reflects the median and box boundaries reflect lower and upper quartiles. In (C), families are shown as points and superfamily proportions as a barplot.

artificial chromosomes (BACs) [26, 49], a majority (75%) of maize LTR retrotransposon families are present only as a single copy in the B73 genome. The average LTR family contains 6.1 copies, with this distribution ranging from 1 to 16,289 copies. In contrast, the family size distribution of TIR transposons is more uniform, with the average family containing 142 elements (range 1 to 9953) and only 10% of families are represented by a single copy. Helitron families are smaller, with 14 copies on average (66% represented by a single copy), and nonLTR retrotransposon families have an average of 3 copies (77% consist of a single copy).

Families are also found nonuniformly along chromosomes (Fig 3B, 3C, 3D and 3E and S3 Fig). Sometimes, the distribution of copies in the largest families in a superfamily match the pattern seen when summarized across all members of a superfamily, such as the five largest RLC families which all share an enrichment on chromosome arms (Fig 3D). There are also families that differ from the aggregate superfamily distribution. For example, the second largest RLG family (RLG00003) is enriched on chromosome arms, and the third largest RLG family (RLG00005) is more uniformly distributed along the chromosome (Fig 3E).

Further, the ages of different TE families vary greatly as well (Fig 4 and S4 Fig). We determine ages of individual copies based on terminal branch lengths of TE phylogenetic trees. For LTR retrotransposons, we additionally measure divergence between the two LTRs of each insertion (See Methods). Some families have not had a new insertion in the last 100 kya, while others have expanded rapidly in that time frame (Fig 4B, 4C, 4D and 4E). Some families display cyclical dynamics, readily generating new insertions that are retained, with pulses of stasis in between (e.g. DTA00073, Fig 4C). Others show sustained activity in the past (e.g. DHH00004, Fig 4B). In total, 70% of TIR families, 20% of LTR families (estimated with LTR-LTR divergence), 15% of nonLTR families and only 7% of Helitron families have been active in the last 100 kya.

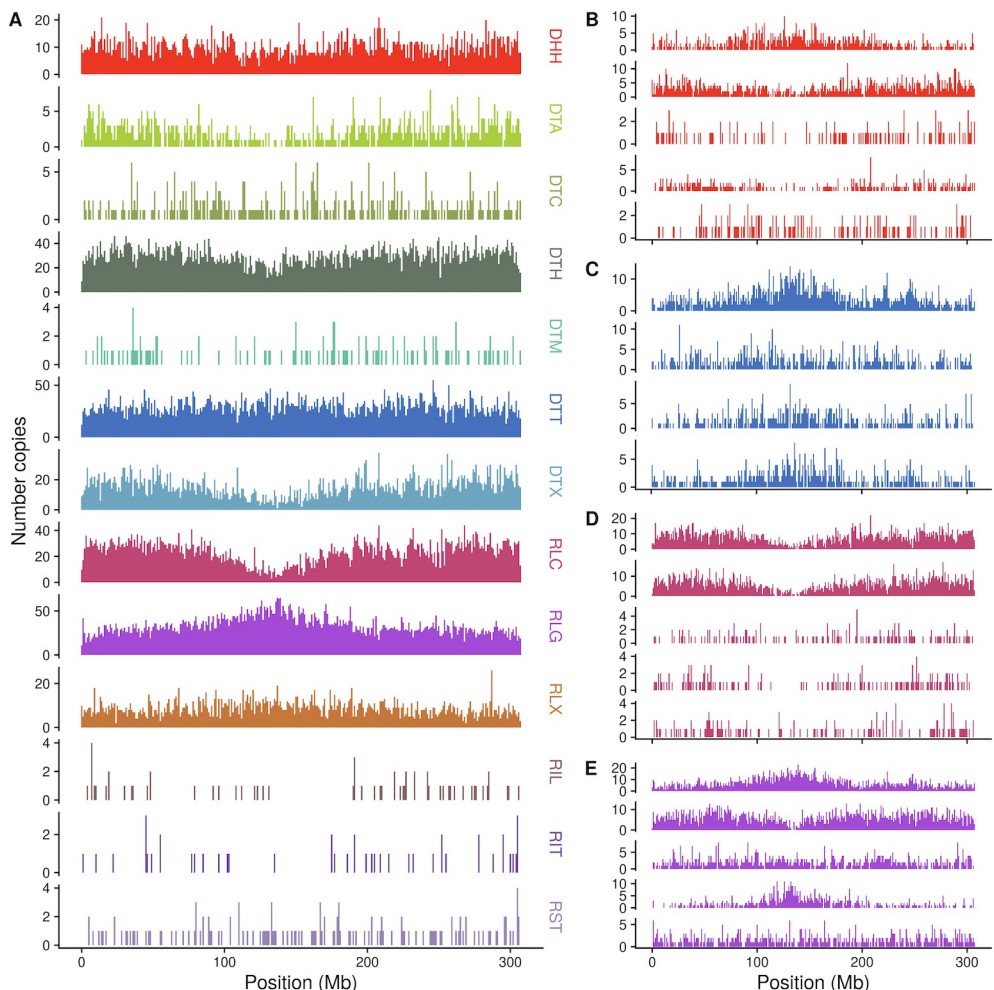

**Fig 3. Chromosomal distribution of superfamilies and example families.** Counts of number of insertions in 1 Mb bins across chromosome 1 for (A) TE superfamilies and (B-E) the 5 families with highest copy number in each of four superfamilies, DHH (B), DTT (C), RLC (D), and RLG (E). Family names are listed in S1 Table.

## Features of the transposition process

Here, we address features that restrict and allow movement of TE copies, as well as influence their survival after insertion.

**TE proteins.** Numerous sequence features of the TE itself are required for the complex transposition process to occur, and these are best understood at the level of TE family. One requirement is the presence of TE encoded proteins that catalyze movement. Functional characterization of TE protein coding capacity is complicated by difficulty in identifying the effect of stop codons or nonsynonymous changes on transposition. Instead we measure homology to TE proteins (see Methods for details), although we recognize this does not fully reflect whether a TE copy can produce a transpositionally-competent protein product. Although TE-encoded proteins are often of similar length within a TE superfamily due to domain conservation and shared ancestry, the longest ORF in a TE varies by family (Fig 5A). Sometimes this is due to the presence of nonautonomous or noncoding copies. While nonautonomous copies rely on protein production in *trans* by other family members, autonomous TE copies encode their

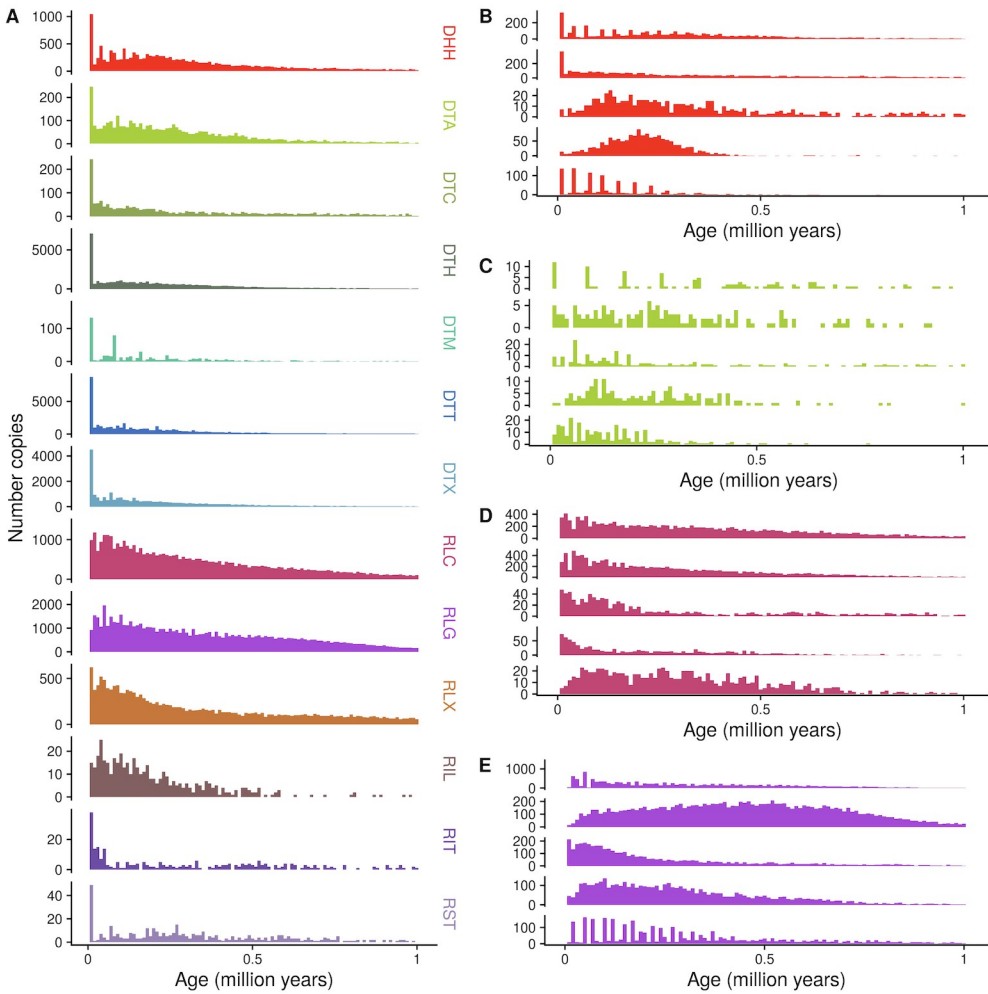

**Fig 4. Age distribution of (A) superfamilies and (B-E) five largest families of (B) DHH, (C) DTA, (D) RLC, and (E) RLG.** Family names are listed in S1 Table. Counts of number of insertions in 10,000 year bins are shown. As they are rare, TE copies older than 1.1 million years are not shown. Ages are calculated with terminal branch lengths for all TEs except LTR retrotransposons, which are calculated with LTR-LTR divergence. See S5 Fig for LTR retrotransposon plots with terminal branch length ages.

own transposition machinery in *cis*. The majority (52%) of LTR families have at least one family member that retains some remnant of coding capacity for all the TE proteins necessary for transposition. In contrast, only 0.6% of TIR families, 0.3% of helitron families, and 0.2% of nonLTR families have at least one family member with protein coding sequence that matches known TE proteins. For all TEs, coding capacity varies substantially between families (Fig 5B). Several LTR retrotransposon families have a small proportion of potentially autonomous copies (S12(C) Fig), and yet other families where coding potential for required proteins is split between between different TE copies (e.g. RLG00001, where 1.6% of copies code only for GAG and 13.5% of copies code for only POL, although both proteins are required for retrotransposition; S12(A), S12(B) and S12(C) Fig). Also, families range from having almost exclusively potentially autonomous coding copies ($\geq$ 75% of copies in 14 families of DTC, RLC, and RLG, S3 Table), to having exclusively nonautonomous noncoding copies (842 families, spanning all 13 superfamilies; S4 Table).

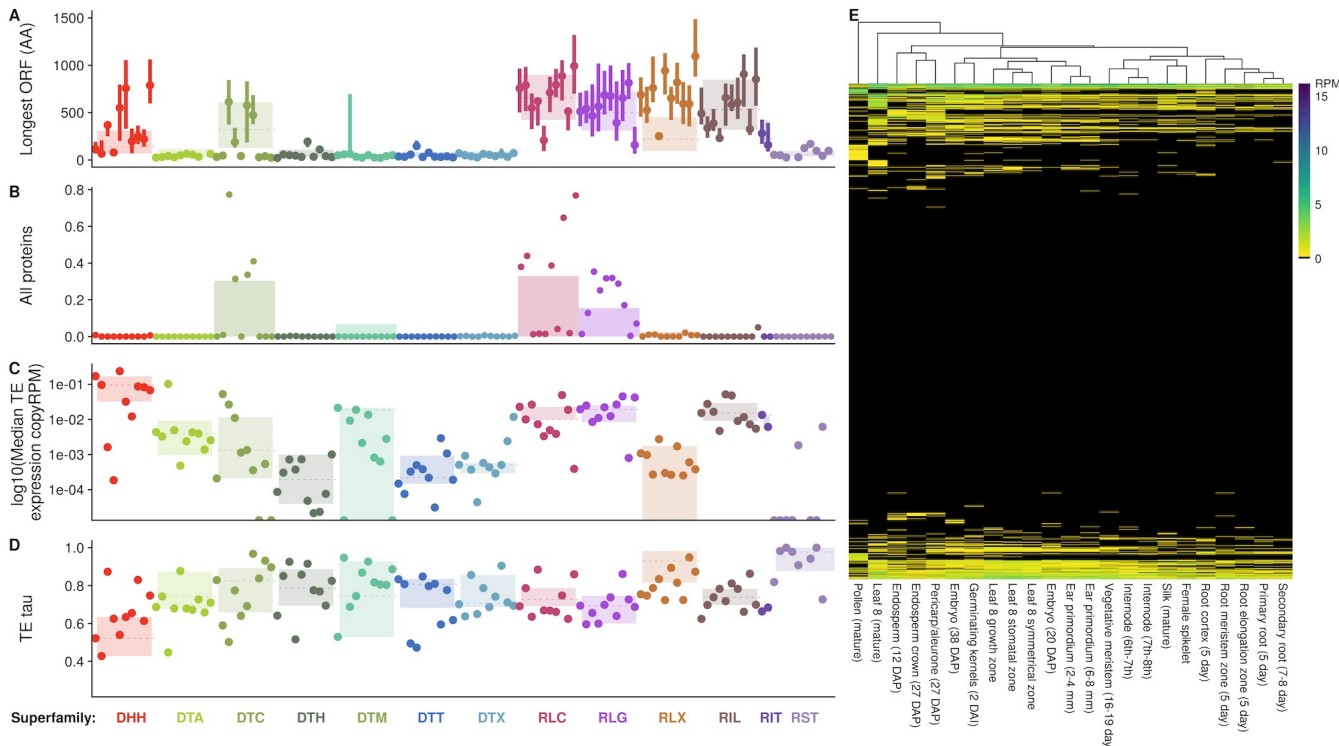

**Fig 5. TEs code for proteins that are expressed, and expression varies by family across tissues.** In A-D, families are in the same order as presented in Fig 2, and listed in S1 Table. (A) Length of longest open reading frame within the TE, measured in amino acids. (B) Proportion of family with all proteins required for transposition. (C) log10 median TE expression across tissues, per-TE copy. (D) Tissue specificity of TE expression $\tau$, with low values representing constitutive expression, and high values representing tissue specificity. (E) Per copy TE expression across tissues (RPM, reads per million), clustered by expression level. Families with greater than 10 copies are shown in rows, and tissues in columns.

Coding capacity for TE proteins likely dictates the ability to generate new insertions, and as such is associated with TE age and the timing of activity of the family. Averaged across all orders, TEs that code for proteins are younger than those that do not code for proteins (median age of 198 kya vs. 263 kya; significant effect of protein coding in Wilcoxon rank sum test, $p < 2e-16$). Further, noncoding copies from families that lack a coding member in B73 show an elevated median age (266 kya) when compared to noncoding copies in families with coding members (174 kya) (all pairwise Wilcoxon rank sum tests show a significant effect of coding status, all $p < 2.1e-11$). For most superfamilies, coding members are older than non-coding copies from families with coding members (S7 Fig).

**TE expression.** Beyond simply coding for TE proteins, another requirement for TE transposition and transgenerational inheritance is expression of the TE itself such that the TE-encoded protein can be generated. Mapping of RNA-seq reads to repetitive TE families is a challenge, as it can be impossible to identify the exact copy that is expressed when a read maps equally well to multiple TE copies [50]. We choose to summarize multiply mapping reads and TE expression at the level of per-copy RPM of the family. This likely averages relevant variation in expression known to exist between copies within maize TE families [51, 52], but reflects patterns observed at the level of the family. Measured in this way, large families are generally transcriptionally repressed (Fig 5C), while small families show higher median per-copy expression levels. Most families are not expressed in any tissues surveyed (Fig 5E). While superfamily medians and median expression per copy of the ten largest families per superfamily show

below 0.1 RPM per copy (Fig 5C), per copy rates of expression can be higher for small families. For example, the 19 copies of RLC00184 (also known as *stonor*) show high median expression of 4.33 RPM per copy.

Tissue specificity can reflect different strategies for TE survival, like that a TE must jump in germline tissue to ensure its transgenerational inheritance at a new locus. Tissue specificity, measured as $\tau$ (see Methods), is highest when values of $\tau$ are equal to 1, and 0 when constitutively expressed at identical levels across all tissues. Helitrons and most LTR retrotransposon superfamilies (RLC and RLG) show lower median $\tau$ than TIR and nonLTR retrotransposon superfamilies (All pairwise Wilcoxon rank sum tests significant ($< 4.2e - 12$) except TIR and nonLTR comparison; Fig 5D). Tissue specificity can be extreme, with some families showing expression in only one tissue (Fig 5D)). For example, DTH00434 shows maximal per copy expression in mature pollen (4.3 RPM), with highly tissue specific expression ($\tau = 0.998$).

**TE regulation.**   TE expression is likely limited by regulation of the TE by the host genome, which we measure via DNA methylation and MNase hypersensitivity in the TE and regions surrounding it. TEs on average are heavily regulated by their host genome: average cytosine methylation across TEs is high (averaged across five tissues, 82% of cytosines in a CG context in a TE are methylated, 67% in a CHG context, and 4% in a CHH context), although this varies across superfamilies (S5 Table) and families (Fig 6A, 6B and 6C). Only a small fraction of base pairs within TEs is in chromatin accessible to MNase, averaging 0.2% in shoot tissue, and 0.08% in root tissue (S9(K) and S9(M) Fig), both lower than genome-wide proportions (0.5% in shoot, 0.2% in root; significantly different from genome-wide values, one-sample Wilcoxon Signed Rank test p $< 2.2e - 16$ for both shoot and root). Despite this overall pattern of regulation, the host genome restricts some families of TEs differently. For example, the median CG methylation of the family DTM00796 is only 52% in anther tissue (Fig 6A), while most other families show higher methylation. There is even more extreme variation in CHG methylation across TE families (Fig 6B): though many TE families show low CHH methylation across the body of the TE, some families of DNA transposons show relatively high CHH methylation (Fig 6C). Although the numbers presented here are for anther tissue, these patterns are robust across tissues (S8 Fig).

Some TEs can preferentially insert to particular genomic locations, often based on local chromatin state [13, 14]. Others can modify the methylation patterns in flanking regions after insertion [32, 33]. Genome-wide, methylation levels in the region surrounding a TE insertion reflect these processes, and are variable both across families and DNA methylation contexts. Of the 1,243 TE families with ten or more copies, median methylation levels averaged across all tissues are elevated within the TE compared to 500 bp away for 734 TE families for CG methylation, 957 for CHG methylation, and 1086 families for CHH methylation. This pattern can be visualized as the decay of methylation moving away from the TE (Fig 6D, 6E and 6F and S8 Fig). The magnitude of reduction in local CG and CHG methylation moving away from the TE differs in extent and pattern, from families where methylation is reduced immediately adjacent to the TE to others with minimal reductions even 2 kb away from the TE (Fig 6D and 6E). In contrast, most families show rapid reductions in CHH methylation within 100 bp away from the edge of the TE (Fig 6F).

**TE base composition.**   Observed DNA methylation levels may be impacted by the base composition of the TE, as cytosines must be present to be methylated. TE families differ in GC content (S9(A) Fig); with extremes ranging from 21% (DTT13542) to 84% (DTH14236) median GC content. This appears to be a consequence of bases carried by the TE itself and not of regional mutation pressure, as variation in GC content in the TE is greater than that of the flanking sequence (S9(B) Fig). For example, GC content in the 1kb flanking DTH14236 is over 30% lower than that in the TE (52% GC in the flanking region). Beyond the proportion of

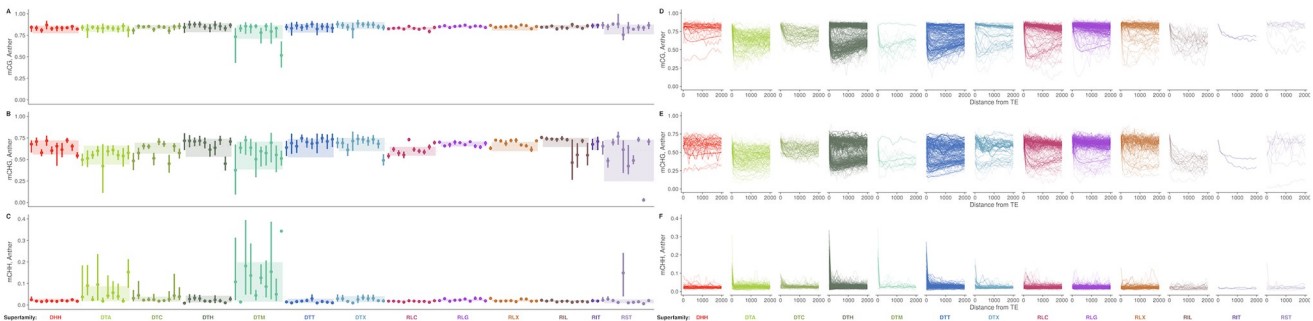

**Fig 6. TEs and their flanking sequences are regulated by their host genome.** Families are presented in the same order as in Fig 2, and listed in S1 Table. CG methylation in TE (A), CHG methylation in TE (B), and CHH methylation in TE (C). CG methylation in 2 kb flanking the TE (D), CHG methylation in 2 kb flanking the TE (E), and CHH methylation in 2 kb flanking the TE (F). All methylation data from anther tissue, other tissues shown in S8 Fig. In (A—C), superfamily median is shown as a dashed line with the interquartile range in the shaded box. In (D—E), median methylation for regions up to 2 kb up and downstream of the TE are plotted for each family, with family size denoted by line transparency (darker lines are larger families).

cytosines in the sequence, the context in which these cytosines are found can impact whether and how they are methylated. For example, 51 families have a median of 0 cytosines that can be methylated in either the CG or CHG context (S6 Table). And even with similar GC content, families differ in the contexts in which they have those cytosines, as families can have moderate GC proportions, but high proportions of these in a CG context (e.g. DTM00473; S9(A) and S9(C) Fig). This is reflected in increased TG content, potentially a consequence of deamination of methylated cytosines (S9(I) and S10 Figs). Notably, TEs with high amounts of methylatable cytosines within the TE do not always share high methylatable cytosine proportions for the region flanking the TE (S9(C), S9(D), S9(E), S9(F), S9(G) and S9(H) Fig).

Although difficulty in mapping short reads to a highly repetitive genome precludes a comprehensive analysis of population frequencies of TEs across maize individuals, we use the proportion of segregating sites within TEs as a proxy for copy number. We measure segregating sites in a panel that includes 1,218 maize and teosinte individuals [53]. While as a whole TEs have fewer segregating sites per base pair (median 0.022) than the genome-wide proportion (0.039) (one-sided Wilcoxon signed rank test, p < 2.2$e$ − 16; S9(O) Fig), some TE families show high numbers of segregating sites (e.g. DTH10060, 0.177 segregating sites per bp). In contrast to the sequence carried by the TE, variation in the region the TE is inserted into is considerably closer to genome-wide averages than that of the TE itself (median 0.034 segregating sites per bp; S9(P) Fig), but still significantly different (one-sided Wilcoxon signed rank test, p < 2.2$e$ − 16).

**Features structuring TE survival after insertion.** The recombinational environment that a TE exists in can impact the efficacy of natural selection on the TE, as higher recombination can unlink deleterious variation from adaptive mutations [54], leading to a positive relationship between recombination and diversity. While LTR retrotransposons are more commonly found in low recombination regions (median 0.30 cM/Mb), Helitrons and TIR elements are more commonly found in higher recombination regions (both show a median 0.43 cM/Mb), and nonLTR retrotransposons are found in the highest recombination regions (median 0.57 cM/Mb) (significant effect of TE order on recombination rate, all pairwise Wilcoxon rank sum tests are significant at p < 1.4$e$ − 06 except the Helitron-TIR comparison). This varies by family—the two largest families of DTT differ in median recombination regions from 0.14 cM/Mb to 0.53 cM/Mb (S11(A) Fig).

Additionally, selection can act on TEs if they have an impact on the expression of genes they land near. Although it is impossible to determine whether a TE insertion causes changes

in nearby gene expression using only the B73 genome, we observe differences among super-families and families of TEs in the expression levels of the closest gene. Across tissues, genes near TIR and nonLTR elements have higher median expression (1.37 RPKM for TIR and 1.83 RPKM for nonLTR) than genes near LTR (1.04 RPKM) and Helitrons (0 RPKM) (S11(C) Fig) (All pairwise Wilcoxon rank sum tests significant, p < 0.023). Notably, this pattern intensifies for genes within 1 kb of the TE, where median gene expression is over 4 RPKM for genes near TIR and nonLTR elements, but 0 RPKM for these genes close to LTR and Helitron elements (S11(D) Fig). Much of this signal is driven by non-syntenic genes—average expression is much higher for the closest syntenic genes (≈12 RPKM) but shows no significant difference amongst orders (Kruskal-Wallis rank sum test, p-value = 0.2353) (S11(F), S11(G) and S11(H) Fig). Some families are often found near highly expressed genes (e.g. DTA00133, median expression 22.38 RPKM), while median expression of the closest gene for $\approx \frac{1}{3}$ of families is close to zero. However, when genes near TEs are expressed, their expression is much more constitutive than that of TE families (S11(E) Fig and Fig 5E), with mean $\tau$ values of 0.75 for genes near TEs and 0.93 for TE families themselves. Tissue specificity varies by family and superfamily as well, and there is a weak correlation between tissue specificity of expression of TE families and expression of the genes they are closest to (Pearson's correlation 0.067, p = 4e-12).

The maize genome arose from an autopolyploidy event [55], and has been sorted into two extant subgenomes [56]. Subgenome A has retained more genes and base pairs than subgenome B, accounting for 64.8% of sequence [48], and 64% of all TEs (S11(B) Fig). Additionally, the median age of TEs in subgenome B is lower (0.24 Mya) than those in subgenome A (0.26 Mya) (Wilcoxon Signed-rank test shows significant effect of subgenome; p < $2.2e - 16$). These differences are likely due to the effect of ongoing transposition erasing any signature of TE differences between parents of the allopolyploidy event, as genome-wide the family with the oldest median age (DTH16531) is only 8.5 million years old.

## Modeling survival of TEs

To account for the myriad differences of these 341,426 TE copies in 27,444 families, we approach our understanding of the survival of TEs in the genome by modeling age as a response to the TE-level features and the genomic regions in which TEs exist today. Age reflects survival of TEs, measuring the amount of time since transposition that they have persisted at a genomic position without being lost to selection or drift. Hence, we measure the predictive ability of features of the TE itself and the genomic region it inserted into on TE survival as measured by age.

Random forest regressions using age as a response variable and features that are measured at the level of the individual TE explain moderate amounts of variance (27.7%), and show low mean squared error (0.014). Across all TEs, information on the superfamily a TE belongs to contributes the most to prediction accuracy for age; after permuting their values, the square root of mean squared error (RMSE) increases by 162 kya (Fig 7B). Other features that increase RMSE by over 100 kya include the size of the family the TE comes from, the length of the TE (both in total bp and when including bases coming from copies nested within it), the TE family, the number of segregating sites per bp within the TE, and the median expression of the TE family across all sampled tissues. In aggregate, features of the region flanking each TE explain approximately as much variation in age as features of the TE itself, but there are more flanking features than those measured on the TE. On average, each feature of the TE contributes over 4 times more predictive power than that of a flanking feature (square root mean squared error of 39 kya for a TE feature, 8 kya for a flanking feature) (Fig 7A and 7B).

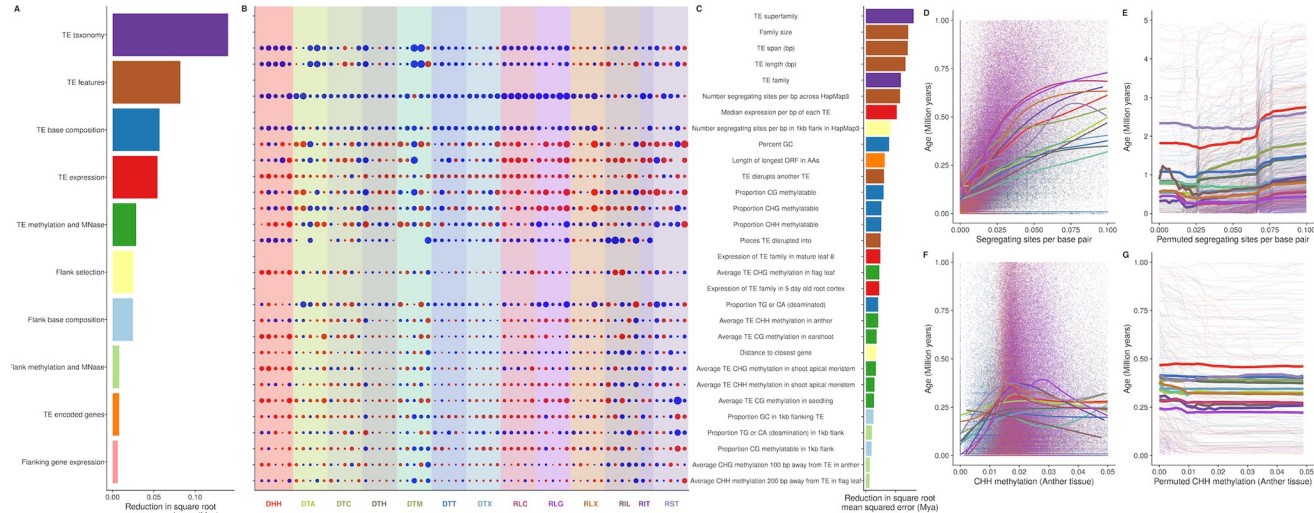

**Fig 7. Features ranked by importance.** (A) Reduction in mean squared error gained by including a feature in a model, summarized into categories. (B) Correlations of each of the top 30 features with age for the five largest families in each superfamily. Features labeled to the right in (C). Size of point is scaled by correlation coefficient, and color by whether the relationship is positive (blue) or negative (red). Rows without values are features that are fixed within a family, thus have no variance. (C) Reduction in mean squared error for top 30 individual features. Colors match categories in (A). (D) Raw correlations between age and segregating sites per base pair (E) Model predictions for the relationship between age and segregating sites per base pair (F) Raw correlations between age and anther CHH methylation of the TE (G) Model predictions for the relationship between age and anther CHH methylation of the TE.

These generalities reflect underlying nonlinearities in the relationships between individual features and age, which are often family-specific. Indeed, correlations of these top features with age differ not only in magnitude, but even in sign between individual families (Fig 7C). To provide additional insight into the local behavior of the relationship between a feature of interest and age, we use the fitted random forest models to predict age for TE copies as we vary the feature. For example, the number of segregating sites in the TE is positively correlated with age in the raw data (Fig 7D), and is confirmed via this permutation approach (Fig 7E). Yet despite this overall pattern, individual families vary in sign (Fig 7C) and slope (Fig 7E) of the relationship. Other features, like CHH methylation of the TE in anther tissue, show relationships that vary by superfamily, where RIT and DHH appear older with increasing CHH methylation of the TE in the anther, while other superfamilies show decreasing age (Fig 7G), a pattern less apparent in the raw correlations (Fig 7F). Across all features, there are largely family-specific combinations of both the direction and strength of correlation with age (Fig 7C). In total, while genomic and TE features contribute to prediction of age, interactions among these features make it difficult to predict the survival of any single family.

## Discussion

### General patterns

As 85% of the maize genome is repetitive sequence [26, 49], and 63% structurally recognizable TE sequence [48], TEs contribute more to the maize genome than sequence that is uniquely 'maize.' Like most plant genomes [11], retrotransposons contribute more base pairs to the maize genome than do DNA transposons (Table 1 and Fig 2B). This is a consequence of the high number of copies (Fig 2A) and the large size of individual retrotransposons (Fig 2C), likely due to a 'copy and paste' replication mode that leaves existing copies intact when

generating new copies. Also like other plant genomes [57, 58], several superfamilies of DNA transposon in the maize genome are found closer to genes than are retrotransposons (Fig 2B). This is likely due to targeted insertion into euchromatic sequences [59, 60], and differences in removal through natural selection after insertion [61, 62].

## TE superfamilies

The bulk of TE sequence is often described at a finer scale, that of individual superfamilies of TEs. Each TE superfamily defined in the maize genome has representatives across the tree of life [63–65], suggesting an ancient origin of these genomic parasites. Some superfamilies have retained dramatic and consistent differences in their spatial patterning across chromosomes over hundreds of millions of years. For example, the superfamily RLG is enriched near centromeres in all plants [66–68] including maize (Fig 3A), highlighting a genomic niche that allows long-term survival near the centromere. Similar patterns exist at deep time scales for DNA transposon superfamilies, which preferentially insert near genes in both monocots and dicots [57, 58, 69, 70] and in maize are enriched on chromosome arms where genes are concentrated (Fig 3A).

These patterns likely reflect the evolution of different ecological strategies of TEs in the genome. Kidwell and Lisch (1997) [7] described two extremes to the 'ecology of the genome'—one, a TE that preferentially inserts far from genes, into low recombination heterochromatic regions, and a second, risky TE that inserts near low copy sequences, more likely to disrupt gene function. We observe these extremes at play in the maize genome, in that LTR retrotransposons dominate the heterochromatic space, with over half of all copies greater than 16 kb from a gene (Fig 2B), and most copies heavily methylated (Fig 6A, 6B and 6C). The alternate strategy also exists in the maize genome, with risky insertions near genes and transcribed regions seen for several TIR superfamilies. For example, over half of Mutator transposons (DTM) are found within 1 kb of a gene (and over one quarter of DTM within 100 bp of a gene) (Fig 2B). This likely results from the preferential insertion of DTM elements upstream of genes [30, 60, 71, 72]. We note that we find TIR copies are found further from genes (17.2 kb) than previously reported for grass genomes [49, 58]. We believe this may be due to previous analyses based on preferential assembly and identification of genic TEs—indeed subsetting to the 893 TIR families found in the Maize TE Database [49] results in a much reduced 1.6 kb median distance to genes. On a genome-wide scale at the level of all TEs, the spatial patterns we observe could result from either preferential insertion or differential removal by selection after insertion. Further characterization of these ecological strategies will be facilitated by investigating TE polymorphism across maize individuals [33, 73] and *de novo* recent insertions that selection has not yet acted on [74].

## TE families

While superfamily level observations are useful for gaining an overview of the distribution and survival of TEs in a genome, more detailed study on a time scale relevant to the evolution of the genus *Zea* comes from studying TE families. Maize TE families are shared with closely related host species, but the number of shared families rapidly decreases with phylogenetic distance. Many families are shared with congeners *Zea diploperennis* [75–77] and *Zea luxurians* [78], but few families investigated are found in maize's sister genus *Tripsacum* (1 mya divergence; [79]) [75–77, 80, 81], and the only families shared between maize and *Sorghum* (12 mya; [55]) are shared only as a result of horizontal transfer events between the species [82]. This suggests that in order to understand TE evolution at a timescale relevant to maize as a

species, it is essential to understand families of TEs, rather than the aggregate properties of superfamilies or orders.

Indeed, our family-level analysis also reveals patterns obscured when TEs are averaged together at the level of superfamily. For example, despite the fact that the RLG superfamily is enriched in centromeric and pericentromeric domains (Fig 3A), the second largest family RLG00003 (homologous to the RLG family *huck* [83]) is predominantly found on chromosome arms (Fig 3D). While many RLG elements contain a chromodomain targeting domain in their polyprotein [84] allowing targeted insertion to centromeres, RLG00003 does not (S12 (G) Fig). This lack of a chromodomain may explain a proximal cause of the observed niche of RLG00003, although other factors are certainly at play, as other families with centromeric enrichment also lack chromodomains (S12(G) Fig). DNA transposons are also best described at the family level. While Mutator (DTM) elements are found a median distance of 2.5 kb from genes (Fig 2B) and have long been observed to target insertions near genes in maize [30, 60, 72], the second largest family, DTM13640, is found a median distance of 34 kb away from genes (Fig 2B). The mechanism for gene targeting seems to be mediated through recognition of open chromatin [60, 71], but precise details of the targeting are unknown. Further investigation into the families that insert near and far from genes may pinpoint how their molecular mechanisms of targeting may differ.

Furthermore, differences in the timing of transpositional activity vary extensively between families. Most TE families in maize have had most new insertions in the last 1 million years (Fig 4). Some TE families have bursts of activity, punctuated by a lack of surviving new insertions, while others appear to be headed towards extinction. All of these timings are much more recent than allopolyploidy in maize ($\approx$ 12 mya) [55] and families show little subgenome bias in their distribution (S9(B) Fig), suggesting that these represent lineages evolving within maize.

Maize was domesticated from teosinte (*Zea mays* subsp. *parviglumis*) 9,000 years ago [85, 86]. It is tempting to address the contribution of TEs to this major transition, especially given the contribution of TE insertions to maize domestication and improvement [87–89]. Although we caution that mutation rates and estimation can complicate ascertainment (see below), 46,949 TEs across all 13 superfamilies have an estimated age of less than 9,000 years, and 24,630 TEs have an estimated age of 0. This suggests that transposition has been ongoing since the divergence of maize from its wild ancestor, but we caution that we lack appropriate confidence intervals for these estimates, especially as non-zero age requires observing at least one nucleotide mutation.

## The family-level ecology of the genome

It can be difficult to predict exactly why a particular TE family differs from other families. Community ecologists aim to understand the environmental factors that give rise to the observed diversity of organisms living in one place, including not just features of the environment but also interactions between species. TE families are analogous to species in the genomic ecosystem, and because the genomic environment a TE experiences is constrained to the cell, TEs are forced to interact in both time (Fig 4) and space (Fig 3). We predict each family of TE is adapted to its genomic ecological niche, where the genomic features we measure represent the environmental conditions and resources limiting a species' ecological niche [90]. TEs additionally can act as ecosystem engineers, modifying the environment they insert into, and generating new habitat for future colonization [10, 44].

In the genomic ecosystem, we can observe interactions between species much like we would see in a traditional ecosystem. We see a number of patterns, including cyclical dynamics

of TE activity through time for several families, sustained activity through time, and a reduction in new copies towards the present (Fig 4 and S4 Fig). This means that the genomic environment a newly inserted TE experiences is affected by the activity and abundance of all other TE families in the genome. At one extreme, members of the same family can even encode different proteins required for retrotransposition in different TE copies, where both types are required to be transcribed and translated for either to transpose. Such a system approaches a mutualism, where the success of one type depends on another. Previous knowledge of these systems was limited to the maize retrotransposon families Cinful, which codes for polyprotein domains, and Zeon, which codes for GAG [25] (represented here by a single family, RLG00001). This strategy has been successful in maize, and RLG00001 [48, 77] for example makes up 135Mb of sequence. *Sorghum*, in contrast, has a genome $\frac{1}{3}$ the size of maize [91] and lacks homologs to RLG00001. Such symbiotic relationships within a TE family have been thought of as remarkably rare [92, 93]; however we identify 25 LTR retrotransposon families where GAG and POL protein domains are found in separate TE copies but less than 1% of copies contain both, suggesting that this pattern is much more prevalent than previously described. These types of elements are best classified as subtypes of a single family, because the *cis* components of the LTR are recognized by protein domains of both GAG and POL proteins, leading to homogenization of sequence signals. As noted by Le Rouzic et al. (2007) [92], symbiotic TE families face a major barrier in being horizontally transferred, as both copies must be transmitted through an already rare process. Their prevalence in the maize genome thus supports instead a long term coevolution of the maize genome and the TEs that live within it, specializing and diversifying with different ecological strategies.

Unlike most contemporary ecological communities, which are censused when a researcher surveys them, the genomic ecosystem carries a record of past transposition. We can investigate this past 'fossil' record using the age of individual TE copies. This allows a robust analysis of the features that define TE survival across time. The TEs we see today are a readout of the joint processes of new transposition—which may not be uniform through time—and removal through selection, deletion, and drift [62]. Survival of a TE can be measured by its age or time since insertion, as our observation of a TE is conditioned on the fact it has not been removed by either neutral processes or selection. Changes in the TE community over time give rise to evolution.

Although relative age differences between TE insertions are limited only by our ability to count mutations, absolute age estimates can be shifted by mutation rate estimates. We use a maize-specific mutation rate [94], which leads to a five times younger estimated age of maize LTR retrotransposons than the 3–6 million years originally estimated by SanMiguel et al. (1998) [95]. Additionally, as nucleotide mutation rates in TEs may be higher than other parts of the genome ($\approx 2$ fold higher in TEs in *Arabidopsis thaliana*, [96]), we consider our age estimates to represent an upper bound of TE age. Nonetheless, age represents a comparable metric of survival in the genome, especially when summarized across multiple copies and families. Our random forest model predicting age of TEs thus relates the action of transposition to the processes that occur afterwards on an evolutionary time scale. The model shows that TE superfamily and family size provide best predictive power for age (Fig 7B).

Another TE feature with high predictive power for age and survival in the genome is the length of the TE, both of itself and its length including copies nested within it. For most families, there is a negative correlation between TE length and age (Fig 7C). However, we find that the relationship between TE length and age in maize is often nuanced, with some long TEs surviving over millions of years (Fig 7C and S13(A) and S3(B) Fig). In other taxa, selection is stronger on long TEs, mediated by a higher potential for nonhomologous (ectopic)

recombination [97–99]. Although a number of factors may contribute to these patterns, it also seems likely that a genome as repetitive and TE-rich as maize perhaps could not have evolved without mechanisms to prevent improper pairing of nonhomologous sequences with high nucleotide similarity [100].

Other predictors of age are expected. For example, we expect a new insertion to be younger if we show that the TE disrupts another TE, which we see for most families shown in Fig 7C. Additionally, we expect the proportion of segregating sites in the TE and the region flanking a TE insertion will be positively associated with TE age, as they reflects a count of the mutations that have accumulated on the haplotype carrying the TE. There is a positive relationship between age and segregating sites for most families shown in Fig 7C. We note that imprecise repair of a double stranded break after excision of a TIR element [70] could obscure this signal to some extent, increasing the number of flanking SNPs while decreasing the average frequency of the TE. Consistent with this mechanism, the superfamily DTT, which excises precisely without introducing nucleotide mutations [101] shows lower median flanking segregating sites per base pair (0.0295) than TIR elements from other superfamilies (0.0310) (Wilcoxon Signed-rank test shows significant effect of DTT superfamily; $p < 2.2e − 16$).

Elevated CHH methylation of TEs has been found in recently activated TEs in *Arabidopsis thaliana* [102] and in TEs near genes in maize [103, 104]. We find complicated, nonlinear relationships of CHH methylation with age (Fig 7F and 7G). These differences between and within families may reflect a natural senescence of TE copies. Young copies not yet silenced by the genome lack CHH methylation, intermediate age copies are effectively silenced with higher CHH methylation levels, and the oldest TEs with low CHH methylation are defunct copies incapable of transposition that are no longer silenced. More detailed study of recently active maize TE families will allow understanding of the temporal dynamics of transcriptional and post-transcriptional silencing of TEs.

In spite of previous predictions, distance to a gene and recombination are not found in the top 30 explanatory variables of age. Old TEs are underrepresented near genes in humans and *Arabidopsis thaliana* [105, 106], consistent with selection against such insertions. Recombination has been implicated in both the removal of TEs and in modifying their impact on fitness via ectopic recombination [6]. We believe that both distance to a gene and recombination rate reflect broad-level summaries of genomic regions, such that they are not predictive in our model once other local features are included. For example, regions with high recombination rate generally show low CG methylation in maize [107], but a subset of genes in such regions show CG methylation across the gene body. Since CG methylation plays a role in TE survival (Fig 7B), inclusion of this feature in our models will thus reduce the importance of recombination rate. Similarly, CHH methylation is most prominent in regions of the genome close to genes, presumably a result of RNA-directed DNA methylation reinforcing the boundary between heterochromatin and euchromatin [103, 104]. As this elevated CHH methylation is often over the TE closest to a gene [104], the distance of a TE to the closest gene may provide largely redundant information beyond what is captured by measurements of CHH methylation. Finally, despite many other features being correlated with either gene density or recombination rate, the two are inextricably linked, as recombination in maize primarily initiates in genes [108]. Together, these combine to reveal few patterns in the relationship between distance to gene, recombination rate, and age of TE copies (S13(C) and S13(D) Fig).

Finally, in spite of the fact our model includes more than 400 features of the genomic environment, TE taxonomy contributes substantially to prediction of TE age (Fig 7A and 7B). We have seen that the relevance and direction of effect of individual features can differ among families (Fig 7C), essentially generating family-specific niches in the genomic ecosystem. In fact, there is no genomic feature we measure which shows even the same direction of

correlation with age across all families. The importance of taxonomy in our model suggests that there are unmeasured latent variables that are best captured with superfamily and family labels. This further emphasizes that the analysis of TEs in maize should focus on family, as each family is surviving in a slightly different way, exploiting a unique genomic niche.

## Conclusion

Genes in the maize genome are 'buried in non-genic DNA' [109] consisting predominately of TEs. The interaction between TEs and the genes of the host genome can structure and inform genome function. The diversity of TEs in an elaborate genome like maize generates a complex ecosystem with many interdependencies and nuances, limiting the ability to predict the functional consequences of a particular TE based only on superfamily or order. Instead, TE families represent a biologically relevant level on which to understand TE evolution, and the features most important for determining survival of individual copies represent dimensions of the ecological niche they inhabit. These observations suggest that the co-evolution between TE and host is ongoing, and inference of the impacts of transposons requires a multifaceted approach. The nuanced understanding generated from exhaustive analysis of genomic features and survival of individual families of TEs serves as a starting point to begin to understand not only TE evolution, but also the evolution of the host genomes they have coevolved with.

## Methods

Scripts for generating summaries from data sources and links to summarized data are available at http://www.github.com/mcstitzer/maize_genomic_ecosystem. Interactive distributions per family can be found at https://mcstitzer.shinyapps.io/maize_te_families/.

### TE sequence properties

We base our analysis on the TE annotation of the maize inbred line B73 [48], updated to more fully capture TIR elements (see S1 Text). TEs that are nested inside of other TEs are divided for further analyses, by assigning each TE base pair in the genome to a single copy by iteratively removing copies in order of arrival. We remove from analysis any TE for which less than 50 bp remains after resolving nested copies. We add the positions of retrotransposon long terminal repeats (LTRs) to these annotations as produced by LTRharvest [110], and delimit the internal protein coding genes of LTR TEs using LTRdigest [111] and GyDb 2.0 retrotransposon gene HMMs [112]. We additionally identify the longest open reading frame (ORF) in each TE model using transdecoder [113], and identify whether this longest ORF is homologous to known transposases, integrases, and replicases respectively for TIRs, nonLTR retrotransposons, and helitrons (JCVI GenProp1044 http://www.jcvi.org/cgi-bin/genome-properties/GenomePropDefinition.cgi?prop_acc=GenProp1044 and PFAM PF02689, PF14214, PF05970) using hmmscan [114] with default parameters. We characterize copies as autonomous based on the content of their protein coding domains, requiring evidence of all 5 proteins (GAG, AP, RT, RNaseH, INT) for LTR retrotransposons, a reverse transcriptase match for LINEs, a transposase profile match for TIR transposons, and a Rep/Hel profile match for Helitrons. This measure is lenient in defining coding content, as it does not penalize stop codons and frameshifts throughout these coding regions.

After insertion, TE copies accumulate nucleotide substitutions that can be used to estimate their age. To estimate age based on divergence of a TE copy from others in the genome, we generated phylogenies of TE copies by first aligning the entire TE sequence of each copy in each superfamily using Mafft [115] (allowing sequences to be reverse complemented with the option `--adjustdirection`) and then building an unrooted tree using FastTree [116].

To make tree building computationally efficient in spite of the high number of TE copies and large element size, we use a maximum of 1000 bp for tree building for the largest 5 superfamilies (3' terminal for Helitrons, 5' terminal for LTR retrotransposons and TIR elements). The terminal branch length of each copy is used as a measure of its age, representing nucleotide substitutions since divergence from the closest related copy in the B73 reference genome. This measure of age makes a number of assumptions about the tempo and mode of transposition—for example, we assume nucleotide mutations in a TE arose at its current location, which may not be true for TIR elements that excise and move to a new location. Nonetheless, it is the only approach to calculate ages of individual TIR and Helitron elements [117, 118] without relying on a consensus element generated from a multiple sequence alignment that can be biased towards recently transposed copies that have not yet been removed by natural selection or genetic drift [118, 119].

Because the 5' and 3' LTR of LTR retrotransposons are identical upon insertion [83], we also estimate their time since insertion using the number of substitutions that occur between the two LTRs. For each LTR retrotransposon copy, we align both LTRs with Mafft [115] and calculate nucleotide divergence with a K2P correction using dna.dist in the ape package of R [120, 121]. For all age measures, we relate nucleotide divergence to absolute time using a mutation rate of $3.3 \times 10^{-8}$ substitutions per site per year [94]. These LTR-LTR estimates are generally in line with terminal branch length age estimates (Spearman's correlation 0.65), with LTR-LTR ages often older than terminal branch length ages (S6 Fig).

## TE environment and regulation

We characterize the genomic environment of the TE and features that overlap the TE. For each TE, we characterize the distance to the closest gene (gene annotation AGPv4, Zm00001d.2, Ensembl Plants v40) irrespective of strand using GenomicRanges [122]. We additionally measure expression of these closest genes across a developmental atlas of the maize inbred line B73 [123] (accessed from MaizeGDB as walley_fpkm.txt using AGPv4 gene names). In order to estimate the overall dynamics and tissue-specificity of expression, we calculated both the median expression and $\tau$ [124] for each of these genes. $\tau$ is calculated as the summed deviance of each tissue from the tissue of maximal expression, divided by total number of tissues minus 1. $\tau$ values thus range from 0 to 1, with low values representing constitutive expression and high values indicating tissue-specific expression. We further characterize whether the closest gene is found in a syntenic position in *Sorghum bicolor* (typically indicative of higher conservation) using curated lists across grass genomes, excluding maize genes matched to multiple *Sorghum* orthologs [125, 126].

In addition to host genes, TEs themselves can be transcribed. Using RNAseq reads from the Walley et al. (2016) [123] expression atlas (NCBI SRP029238), we counted reads that align uniquely to a specific member of a TE family, as well as multiply mapped reads that align to a single family, as in Anderson et al. (2018) [51] and Anderson et al. (2019) [52]. This allows estimation of the expression level of a TE family, despite the repetitive nature of TEs that limits unique mapping of reads. Reads that map to TEs located within genic sequences (generally within introns) were excluded because their expression is indistinguishable from transcription from the gene promoter. We take the mean value of reads per million across the two to three replicates per tissue, and divide by the total family size to get a per-copy metric of expression. As with genes, we calculate median expression across tissues and tissue specificity using $\tau$.

To identify the recombinational environment in which each TE exists, we use a 0.2 cM genetic map of maize generated from the Nested Association Mapping (NAM) panel [127]. We convert AGPv2 coordinates to AGPv4 coordinates using the Ensembl variant converter

[128]. To approximate the recombination rate in genomic regions, we fit a monotonic polynomial function to each chromosome [129]. Using this function and TE start and end positions, we calculate a cM value for each TE, and convert to cM/Mb values by dividing by the length of the TE in megabases.

The chromatin environment a TE exists in can impact transposition [60]. We converted data on MNase hypersensitive sites in roots and shoots [130] from the AGPv3 reference genome to AGPv4 coordinates using the Ensembl variant converter [128]. We counted how many hypersensitive sites exist in each TE, as well as the proportion of base pairs of the TE that are hypersensitive. We also calculate these metrics for the 1 kb region flanking the TE on both sides.

Regulation of TEs by the host genome is often mediated via epigenetic modifications. We use bisulfite sequencing reads from shoot apical meristem, anther, ear shoot, seedling leaf, and flag leaf [104, 131]. We trim adapters using TrimGalore, map using bsmap 2.7.4 with parameters (`-v 5 -r 0 -q 20`) [132], and summarize in 100 bp windows as in Li et al. (2015) [104], to characterize the local proportion of methylated cytosines in all three contexts (CG, CHG, CHH; where H is any base but G). We summarize the average levels of each measure over each TE copy and each of 20 100 bp windows of flanking sequence on either side, imputing missing data with the family mean.

To identify differences between TE copies in their base composition, we calculate GC content plus the number of di- and tri- nucleotide sites containing cytosines in a methylatable context (CG, CHG, CHH). We count these contexts in each TE using the bedtools `nuc` command [133] and divide by TE length to determine the proportion of the sequence that is methylatable for each context. We also calculate these measures of methylatability for the 1 kb flanking the TE on each side.

We also measure the number of segregating sites per TE base pair and the 1 kb flanking in the *Zea mays* Hapmap3.2.1 dataset [53] as well as the subgenome [48] each TE is found within.

As we cannot calculate accurate summaries of genomic features for families with a small number of TE copies, we include only those families with more than ten copies when presenting results in the text that identify specific outlier families, such as the family with highest GC content. When presenting summaries at the superfamily and order level or results modeling TE age, we include information from all TE copies, including those from smaller families.

## Analysis and interpretation

We implement random forest regression models (in the R package 'randomForest' [134]) to understand the importance of different genomic features to TE survival in the genome as measured as the age of individual extant copies. We train models on 50% of copies, and summarize 1000 iterations of trees. The remaining TEs are retained as a test set to estimate model performance. Any missing data is assigned a value of -1, and the categorical variable of superfamily is considered as a factor. Because of limitations in converting numbers to binary, we limit the categorical variable of family to the 31 largest families, and code all others as 'smaller.' We summarize the overall importance of each feature in predicting age by permuting its values across individual TE copies and observing the change in mean squared error of the model prediction of the actual value, scaled by its standard deviation. We summarize features into categories reflecting TE taxonomy, TE base composition, TE methylation and chromatin accessibility, TE expression, TE-encoded proteins, nearest gene expression, regional base composition, regional methylation and chromatin accessibility, and regional recombination and selection. A full description of the individual measurements that go into each category is found in S2 Table.

In order to interpret family-specific relationships for the top predictors of age, we perform further analyses. We calculate the Pearson's correlation coefficient of each predictor with age, using samples from each family. To visualize the nonlinear relationships and interactions produced by such models, we calculate Individual Conditional Expectations (ICE plots [135], R package 'pdp' [136]), which summarize the contributions of permuted values of a variable of interest to the response, while conditioning on observed values at all other variables. We provide permuted values summarizing 95% of the observed data, to provide predictions in a region of parameter space the model is trained on. We summarize these responses as deviation of the predicted value generated with permuted data from the true value, and plot as individual lines and superfamily averages.

## Supporting information

**S1 Fig. Family characteristics of each of the largest 10 families of each superfamily with at least 10 copies.** (A) Proportion of TEs within the transcript of a gene, including introns and UTRs. (B) TE span along the genome, summing both the base pairs of the TE and the base pairs of the TEs nested within it. (C) Proportion of TEs that are intact, that is, uninterrupted by the insertion of another TE. In (A and C), families are shown as points and superfamily proportions as a barplot, and in (B) families are shown with medians as points and lines representing ranges of upper to lower quartiles, with superfamilies shown as colored rectangles.
(TIF)

**S2 Fig. Chromosomal distribution of superfamilies across all 10 maize chromosomes.** Count of TE copies of each superfamily in 1 megabase bins across each chromosome.
(TIF)

**S3 Fig. Distribution on chromosome 1 of five largest families with at least ten copies in each superfamily.** Count of TE copies in 1 megabase bins along chromosome 1. (A) DHH, (B) DTA, (C) DTC, (D) DTH, (E) DTM, (F) DTT, (G) DTX, (H) RLC, (I) RLG, (J) RLX, (K) RIL, (L) RIT, (M) RST. Note that some families have no copies on chromosome 1, including DTT10880 and DTX10177. Additionally, the RIT superfamily only has two families.
(TIF)

**S4 Fig. Ages in 10,000 year bins across each of the largest 10 families of each superfamily with at least 10 copies.** (A) DHH, (B) DTA, (C) DTC, (D) DTH, (E) DTM, (F) DTT, (G) DTX, (H) RLC, (I) RLG, (J) RLX, (K) RIT, (L) RIL, (M) RST. The RIT superfamily only contains two families.
(TIF)

**S5 Fig. LTR-LTR ages and terminal branch length ages for LTR retrotransposons.** Ages in 10,000 year bins across each of the largest 10 families of each superfamily with at least 10 copies. Left plots (A-D) show LTR-LTR ages, right plots (E-H) show terminal branch length (TBL) ages. (A) all copies, LTR-LTR, (B) RLC families, LTR-LTR, (C) RLG families, LTR-LTR, (D) RLX families, LTR-LTR, (E) all copies, TBL, (F) RLC families, TBL, (G) RLG families, TBL, (H) RLX families, TBL.
(TIF)

**S6 Fig. LTR-LTR ages vs. terminal branch length ages for LTR retrotransposon superfamilies.** Spearman's correlation coefficient shown on plot for each superfamily.
(TIF)

**S7 Fig. Age of TE copies split by coding potential of self and family.** Violin plots with three lines, at median and 25th and 75th percentile. Only ages younger than 2 million years are shown. "Coding copy" refers to those copies that code for protein, "noncoding copy" refers to those copies that don't code for protein, but a family member does, and "noncoding family" refers to copies from families without a coding member in B73.
(TIF)

**S8 Fig. Methylation in TE and flanking sequence, across tissues.** A-J: mCG; K-T: mCHG; U-end mCHH. Tissues on y-axis, from top to bottom: Anther, SAM (shoot apical meristem), Ear-shoot, Flagleaf, Seedling leaf.
(TIF)

**S9 Fig. Features of the TE and flanking sequences.** GC content in the TE (A) and 1kb flanking sequence (B). Proportion of sites methylatable in the CG context in the TE (C) and 1kb flanking sequence (D), methylatable in the CHG context in the TE (E) and 1kb flanking sequence (F), proportion of sites methylatable in the CHH context in the TE (G) and 1kb flanking sequence (H). Proportion of sites containing a TG or CA dinucleotide in the TE (I) and 1kb flanking sequence (J). Proportion of sites in MNase hypersensitive regions in root in TE (K) and 1kb flank (L), and shoot in TE (M) and 1kb flank (N). Proportion of segregating sites in the TE (O) and 1kb flank (P).
(TIF)

**S10 Fig. The proportion of methylatable cytosines is negatively correlated with the proportion of TG/CA dinucleotides.** The x-axis reflects the proportion of cytosines in a CG context within the TE, and the y-axis reflects the proportion of dinucleotides in the TE that contain a TG or CA.
(TIF)

**S11 Fig. Recombination, subgenome, and expression of closest gene.** (A) Recombination rate across the TE, (B) proportion of TEs in subgenome A, (C) log10 median expression of the closest gene to each TE, (D) log10 median expression of genes within 1kb of the TE, (E) Tau of closest gene to each TE, (F) log10 median expression of the closest syntenic gene, (G) log10 median expression of closest syntenic genes within 1 kb, and (H) Tau of the closest syntenic gene.
(TIF)

**S12 Fig. Protein coding gene presence of individual LTR GAG and POL domains.** Shown are (A) the proportion of TEs with evidence of agglutination factor (GAG) domain present, (B) the proportion of TEs with evidence of all polyprotein domains present (aspartic protein-ase, integrase, reverse transcriptase, and RNaseH), (C) the proportion of TEs with both GAG and Polyprotein present in the same element. Families are shown as points and superfamily proportions as barplot.
(TIF)

**S13 Fig. Predicted and observed relationship of age to TE length and distance to gene.** Raw relationship (A & C) and predicted relationship (B & D) of TE length (A & B) and distance to gene (C & D).
(TIF)

**S1 Table. Ten largest families in each superfamily, as shown left to right in plots.**
(TXT)

**S2 Table. Categories that each feature measured for each TE is classified into.**
(TXT)

**S3 Table. 14 families with at least 10 copies in the B73 genome, with at least 75% of copies coding for transposition related proteins.**
(TXT)

**S4 Table. 842 families with at least 10 copies in the B73 genome that lack coding representatives.**
(TXT)

**S5 Table. Mean methylation levels across superfamilies, averaged across all tissues, and averaged within a tissue.**
(TXT)

**S6 Table. TE families that lack methylatable cytosines (presented as family median values).**
(TXT)

**S1 Text. TIR annotation methods.**
(PDF)

## Acknowledgments

This work was inspired by the concept of transposable elements exisiting within "niches in the ecology of the genome," as introduced by Margaret Kidwell and Damon Lisch [7].

## Author Contributions

**Conceptualization:** Michelle C. Stitzer, Sarah N. Anderson, Nathan M. Springer, Jeffrey Ross-Ibarra.

**Data curation:** Michelle C. Stitzer, Sarah N. Anderson.

**Formal analysis:** Michelle C. Stitzer, Sarah N. Anderson.

**Funding acquisition:** Michelle C. Stitzer, Jeffrey Ross-Ibarra.

**Methodology:** Michelle C. Stitzer, Sarah N. Anderson.

**Software:** Michelle C. Stitzer, Sarah N. Anderson.

**Supervision:** Nathan M. Springer, Jeffrey Ross-Ibarra.

**Visualization:** Michelle C. Stitzer.

**Writing – original draft:** Michelle C. Stitzer.

**Writing – review & editing:** Michelle C. Stitzer, Sarah N. Anderson, Nathan M. Springer, Jeffrey Ross-Ibarra.

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
