## [Decision Letter · Decision Letter 0]

17 Sep 2019

Dear Dr Stitzer,

Thank you very much for submitting your Research Article entitled 'The Genomic Ecosystem of Transposable Elements in Maize' to PLOS Genetics. Your manuscript was fully evaluated at the editorial level and by independent peer reviewers. The reviewers appreciated the attention to an important problem, but raised some substantial concerns about the current manuscript. Based on the reviews, we will not be able to accept this version of the manuscript, but we would be willing to review again a much-revised version. We cannot, of course, promise publication at that time.

Should you decide to revise the manuscript for further consideration here, your revisions should address the specific points made by each reviewer. In particular, the revision should address the specific comments of Reviewer 1 regarding clarity of figures, description of transposable element taxonomic categories for the general reader, and statistical support for some assertions. Additionally, both reviewers point out that the "genome ecology" concept, while intriguing, is largely employed in this manuscript as an analogy rather than a mode of inference. This should be addressed in a revised manuscript. We will also require a detailed list of your responses to the review comments and a description of the changes you have made in the manuscript.

If you decide to revise the manuscript for further consideration at PLOS Genetics, please aim to resubmit within the next 60 days, unless it will take extra time to address the concerns of the reviewers, in which case we would appreciate an expected resubmission date by email to plosgenetics@plos.org.

[LINK]

We are sorry that we cannot be more positive about your manuscript at this stage. Please do not hesitate to contact us if you have any concerns or questions.

Yours sincerely,

Jesse Hollister

Guest Editor

PLOS Genetics

Kirsten Bomblies

Section Editor: Evolution

PLOS Genetics

Reviewer's Responses to Questions

**Comments to the Authors:**

Reviewer #1: This is a “bird’s eye” view of transposons in the maize genome. It makes the assertion that the maize genome constitutes an ecosystem, in which different families of elements occupy distinct niches. Certainly, the shear diversity of elements with respect to their location, methylation and expression supports this idea. I’m not sure the authors really support the idea that elements actually interact with each other, which weakens the argument. For instance, evidence that one family actually negatively effects the activity of another family would support the hypothesis, as would evidence that some elements tend to insert into other elements rather than themselves. However, evidence of cooperation between families (Cinful and Zeon) is intriguing. Based on the data supplied here, I would suggest that the genome ecology hypothesis is, at this point, mostly a useful metaphor, and a good place to start for further analysis. As indicated below, I think presentation of the data in the figures could be improved for greater clarity. If I was confused, I’m guessing other readers will be confused as well). I would have also liked to have the definitions of superfamilies and families more precisely defined, and I would have liked to have had instance in which the variation observed really does support and ecological model. Overall, because the subject is so broad (hard to believe when we are looking at a single genome!), I think the authors did a pretty good job, particularly since all of the raw data and code has been made available. I very much hope that this information is integrated into TE models in the current maize genome version, since it’s a mess right now. Each section could (and likely will be) a manuscript on its own, but this manuscript opens doors for numerous other, more in depth analyses.

Line 28. That’s a pretty big claim. The manuscript should support it, but I’m not sure it does.

Line 41. A quibble, but maybe say maize genome?

Line 93. As far as I know, this is still a hypothetical mechanism for eukaryotic Helitrons, since no one has demonstrated this mechanism in these organsism. Best to say, “are thought to...”

Line 176. It might be useful to remind the reader as to how orders and superfamilies are defined. Just a couple of sentences for those not familiar with the terminology.

Line 190. Typo.

Line 209. Not sure it’s worth doing, but it would be interesting to know if some TEs are more attractive targets for insertion - this would fit into a competition model. For instance, do some TEs avoid insertions into themselves? That would be kind of cool.

Line 213. Again, it sure would be interesting to see if there are any biases in insertions. I’m not trying to make more work for the authors, but this would be really interesting. Also, give my admittedly anecdotal evidence, I’m surprised that these TIR elements are not closer to genes.

Figure 2. For the non-specialist, the names in the figure don’t mean much, so it is hard for them to get the major point, other than length of different TEs varies. Perhaps it would be worth adding something in the legend saying, “Element designations beginning in a DT represent TIR DNA elements, those beginning in a DH are helitrons,” and so on. The same perhaps in the other Figures with these designations? I know it’s redundant, but I think it’s always a good idea to keep the reader oriented, so that if they see an exceptional family, they will know right away what kind of element it is.

Figure 3. It is very difficult to match the color designations below with the graphics above. In addition to the label, I would go ahead and individually label each tract (particularly for color blind readers, this would be a big help).

Line 219. Again, a reasonable claim is made concerning length, but you could do the math and see if that is the case. That is, does is the length of elements the sole determinant of whether or not it has an insertion into it?

Line 233. “Arrived in the genome?” Does that mean that they were horizontally transferred from another species? Do you mean that a given copy of these elements at a given position is 350,000 years ago?

Line 249. Previous analysis has suggested that the average distance of a TSS to a TE is roughly 300 bp (with a lot a variation). On net, these distances seem much farther. Do you have a calculation of the average distance upstream of TSSs of TEs?

Line 254. It might be useful to remind the reader as to how families are defined.

Line 289. “estimated with LTR-LTR” should be “estimated using LTR-LTR”. Also, if you are going to say this, then how are the age of the other elements established? I’m assuming by divergence from the consensus sequence of a given family. Perhaps it would be a good idea to reference your materials and methods here, since dating TEs is a non-trivial task.

Line 294. Well, yes and no, since you are looking at a combination of any restrictions and allowances and selection following insertion. It’s important that this distinction is made clear.

Line 302. Which proteins? Those that are known to be functional (hAT, CACTA and MULEs have functional transposases) and proteins that are full length and are potentially functional?

Line 315. Confusing wording. 0.6% of families within a super family (implied by the wording), or 0.6% of elements within each family?

Line 319. Should read “potentially autonomous elements”. Autonomy is a functional designation.

Figure 5. As with the other figures, this figure could be more clear. In 5A and B, since each dot has a range around it, I’m assuming there are multiple proteins represented. Are these families (i.e. RLG0001)? And for each color, all of the families within each superfamily that encode a protein are shown? I get it now, but I think the more explicit you can be the better. In 5C, “Presence of GAG, all five domains (GAG and Pol), and Pol (which encodes four domains) in LTR retrotransposons.” So, GAG has five domains that includes a Pol, and Pol (a different Pol?). A bit confusing. Why are all the superfamilies listed with and without colors? What is the point of the list in which they are black? In 5G, it’s not clear how the two axes are organized. Is the X axis organized by similarity of patterns of expression? If so, how is the Y axis organized? The color bar on the left seems to suggest that patterns of expression are more or less randomly distributed, but other than that, I’m not sure how informative this is. And what is “sup”?

Line 354. Maybe or maybe not, since you are dividing the expression level by the total copy number, so higher copy number elements will inevitably show lower per element expression levels. It’s certainly possible that a small subset of a large family is expressing at a relatively high level. It might have been useful to only look at RNAseq data that maps perfectly to intact elements (good tsds, good ORFs), but there are problems with this as well.

Line 372. It is well known that tissue specificity in pollen is likely due to transient relaxation of silencing in the vegetative nucleus and is related to potential activity rather than actual activity. The tough part of this analysis is TEs are often transiently expressed in response to stresses, and which is not represented in this panel. Actually, if data existed for stressed plants in a subset of tissues, you might have gotten some interesting data. As it is, I think the most you can say is that most TEs, under normal conditions, are probably expressed at very low levels.

Line 397. Did the level of methylation correlate with expression level? Also, it is worth pointing out the CHH methylation is largely restricted to TEs adjacent to genes, so position, rather that the type of element can be important. If a given family (e.g. MULEs) are closer to genes, they are more likely to have CHH methylation. This may or may not have to do with how effectively they are silenced.

Line 399, 400. typos.

Line 399. I don’t know what this means. Preferential insertion based on methylation?

Line 400. Or that they insert into regions that are already methylated. How many of the flanking sequences are methylated because they are TEs? For many of these flanking sequences, the only way to know for sure if the TE is causing an effect is if it is polymorphic. I believe the W22 methylome and sequence is available; a comparison would have been illuminating.

Line 415. Yes, because these are TEs that nucleate CHH islands.

Figure 6. Now the colors are no longer designated at all. Here and throughout, I would indicate right in the figure, above the data, what the superfamilies are. Also, since the point of this figure is, in general, to compare the TE and the flanking sequences, why not put B directly under A so that a visual comparison is possible. Frankly, although I accept the overall conclusions, with this and other figures I had trouble getting the point, other than that different families have differences in methylation in the TEs and their flanking sequences.

Line 413. This certainly seems to be true for some, but certainly not for all. Here and throughout some statistical support for what seems visually apparent would be helpful. Also, were the flanking sequences filtered for being TEs or not? What would TEs inserted into other TEs look like? What would families that are more likely to be inserted into other TEs look like.

Line 431. Are these older elements where there was a lot of C to T conversion due to methylation?

Line 477. Statistical support for a difference?

Line 486. The maize genome is notorious for having large numbers of genes that are unlikely to actually be genes. I wonder what this analysis would look like if only syntenic genes were examined. I think this would be more informative.

Line 494. This kind of makes sense, since it is likely that some elements insert into regions of relatively open chromatin and are thus more likely to be expressed.

Line 504. Interesting. Statistically significant?

Line 536. What about the effects of C to T mutations due to methylation?

Line 565. Is this because there are more likely to be more polymorphisms?

Line 571. This is surprising, given what we know about selection against TEs near genes and the fact that CHH is mostly associated with TEs near genes.

Line 616. There are better, more recent references for this.

Line 631. This is reasonable, but what if selection acts more efficiently to remove large LTR insertions near genes?

Line 633. This is also supported by de novo insertion data.

Line 650. That is fascinating. I had no idea that families diversified so quickly. Then again, this depends on how families are defined.

Line 695. Awkward wording.

Line 726. Do you mean newly inserted?

Line 732. Any evidence for that?

Line 738. Any evidence that Zeon and Cinful work together? This is an intriguing hypothesis, but it is speculative and should presented as such. Still, this a really intriguing idea!

Line 779. I’m not clear as to why this is not a somewhat trivial observation.

Line 813. As is evidenced by the very low recombination rates outside of genes in maize.

Reviewer #2: This paper provides a detailed, rigorous description of the properties, abundances, distributions, and ages of transposable elements in the maize genome. The analyses are impressive and, in general, the paper is well-written (but see below for some significant issues). I think it will make a valuable contribution to the literature. However, I do have some comments for revision, which I present in the spirit of improving an already strong paper before publication.

Major points:

- I think the data presented in this paper are very useful, and the paper provides a very detailed overview of TE content in the maize genome. However, I struggle to see just how this is an example of "genome ecology" properly understood, for two reasons:

1) The paper does not actually use any methods from ecology. So, here -- as in most previous uses of the concept -- "ecology" serves as analogy rather than analysis. It's a bioinformatics study that describes TE properties and then layers on the ecological analogy. This is fine, but it may be worth noting that this is what is going on as opposed to, say, actually using ecological methods with genome data.

Examples of studies that are explicitly "genome ecology" in methodology and not just metaphor include:

Saylor et al. (2013). A novel application of ecological analyses to assess transposable element distributions in the genome of the domestic cow, Bos taurus. Genome 56: 521-533.

Serra et al. (2013). Neutral theory predicts the relative abundance and diversity of genetic elements in a broad array of eukaryotic genomes. PLOS One 8: E63915.

Linquist et al. (2015). Applying ecological models to communities of genetic elements: the case of neutral theory. Molecular Ecology 24: 3232-3242.

2) As noted by Linquist et al. (2013; Bio. Rev. 8: 573-584), most previous examples of "genome ecology" were actually "genome evolution". They used population genetics models or phylogenetics, focused on evolutionary timescales, involves evolution of TEs and genomes, and so on. This paper actually tends to drift into this category as well. Notably, most of the section on "The Family-level Ecology of the Genome" talks about evolutionary mechanisms and patterns; there is actually very little ecology discussed.

- The Introduction is very well written, and it is refreshing to see a nuanced overview of TEs rather than the trope of "long dismissed as junk DNA...". Where the paper needs work is in the Results and Discussion. Here, the paper suffers from what is sometimes called "thesis syndrome", where a paper for publication reads very much like a graduate thesis chapter.

1) Results -- This section contains way too much material that should be in the Discussion (or Introduction). Interpretations of findings, reviews of previous literature, etc., do not belong in the Results section. Here, the findings should be presented without interpretation, allowing the reader to agree with or challenge the patterns reported independently of whether they agree with your interpretation.

2) Discussion -- This section spends far too much time discussing the work of others. The Discussion in a data paper (versus a review paper) should discuss your results first and foremost. Citing the literature should mostly be done to put your findings in broader context (after you have discussed them, not "so and so said... and we found that too") and/or to back up an interpretation that you wish to present or a claim you are making. Major sections of the Discussion are more like a review paper than a discussion of the results in a data paper.

Minor comments:

- Despite what elementary school students are taught, there is no specific grammatical rule that forbids the use of "But" to start a sentence. However, doing so 10 times in a single paper is excessive and inelegant (the tendency to over-use it is probably why elementary school students are told not to do it at all).

- p.7 -- "(Figure 2; Interactive distributions per family: )." Not sure what this is supposed to say.

**Have all data underlying the figures and results presented in the manuscript been provided?**

Reviewer #1: Yes

Reviewer #2: Yes

PLOS authors have the option to publish the peer review history of their article (what does this mean?). If published, this will include your full peer review and any attached files.

Reviewer #1: No

Reviewer #2: No

---

## [Decision Letter · Decision Letter 1]

13 Jul 2020

Dear Dr Stitzer,

Thank you very much for submitting your Research Article entitled 'The Genomic Ecosystem of Transposable Elements in Maize' to PLOS Genetics. Your manuscript was fully evaluated at the editorial level and by independent peer reviewers. The reviewers appreciated the attention to an important topic but identified some aspects of the manuscript that should be improved.

We therefore ask you to modify the manuscript according to the review recommendations before we can consider your manuscript for acceptance. Your revisions should address the specific points made by each reviewer.

[LINK]

Yours sincerely,

Kirsten Bomblies

Section Editor: Evolution

PLOS Genetics

Kirsten Bomblies

Section Editor: Evolution

PLOS Genetics

Thanks for your edits! The paper is much improved. The reviewer however makes some remaining comments and I think that some of these are worth addressing. So I am giving this a "minor revision" - it will not need to go back out for review after these changes. Thanks!

Reviewer's Responses to Questions

**Comments to the Authors:**

Reviewer #1: Please note that I think the manuscript is fine as it is (although I do think some aspects of figure 7 could be clear). My comments below are simply meant to provide food for thought.

Line 176. I think I get what the authors are getting at here, but it occurs to me from the text here, as well as the other reviewers comments, that I’m actually not entirely clear as to the distinction between ecological and evolutionary processes. That’s probably just because I’m not a specialist in either field, but what exactly is the distinction. In this context, does “ecological” mean the interaction between different TEs and between TEs and their hosts? I’m assuming here that it does not mean the affects that TEs have on ecological processes occurring at the level of the host. Does it also mean interactions that have an impact on survival of a given TE lineage? A google definition (sorry, I mentioned that I’m not a specialist) of niche is “all of the interactions of a species with the other members of its community, including competition, predation, parasitism, and mutualism. A variety of abiotic factors, such as soil type and climate, also define a species’ niche.” What are the analogies here? Taking a gene-centric view, I suppose each group of TEs that can cross mobilize constitute a “species”, and TEs “speciate” when one lineage gives rise to two lineages that can no longer cross mobilize. Individuals would be individual stretches of DNA that can replicate (TEs), or can contribute to host function (host genes). However, with TEs get more squirrely, since any give insertion is, in situ, “alive” only to the extent that it retains the capacity to replicate, which is a function of its particular niche. Or something like that. Ecological functions would have to do with all of the particular features that would allow a given TE (or, more properly, TE lineage) to thrive over time, as well as the impact that this TE has on both the other TEs and on host fitness. In contrast, Evolutionary process would be what? Time scale? Neutral processes that result in large changes in genome architecture that are random with respect to ecological processes (at the genome level)? Would this be an example of the distinction? Some TEs target heterochromatin (and thus often other TEs) and some targe TSSs by tethering to components of the POLII complex. These two distinct strategies represent “niches”, with both costs a benefits, so this is Ecological. In contrast, Gout has shown that selection over time tends to remove TEs from regions immediately adjacent to genes, so TEs near genes tend to be younger. So if you have one TE family that is older than a second family, the older family will be distributed differently than the younger family even if the TEs have exactly the same targeting initially. These TEs do not occupy distinct “niches”; the differences are purely a function of selection at the level of the host and time. So in figure 3, if age were mapped on to these distributions, you might find a correlation between age and presence in the pericentromeres (CACTAs are, as I recall older and found more frequently in these regions). Similarly, one can imagine a variety of stochastic events that ends up favoring a given TE family that have nothing to do with particular strategies employed by a given TE that impact distribution. Horizontal transfer, for example, that moves a TE from a highly suppressive environment to a more permissive environment. These would be “Evolutionary” processes. Please forgive me for going on, but I’m just trying to give the authors a sense of what comes up for me when reading the text. Hopefully it is of some use to the authors.

Line 208. Thank you for not using the word “respectively” : )

Line 246. Is “years ago” correct? Is the median age of people in nursing homes 82 years ago?

Line 267. Good call. Since plant genomes have such a high turnover rate, even if a given lineage did “arrive in the genome” a very long time ago, we could not know when.

Line 299. This makes sense, as the copy number of older families, and the overall number of families as defined would go up as time passed. So if you started with a massive bloom and 10,000 copies, all of which were DOA, then, as copies were deleted and the sequences diverged to about 80% similar, you would end up with hundreds of distinct “families”, each with a very low copy number.

Line 303. Both the TIR and helitron examples make me think about gene capture. In rice, there are about 3000 Pack Mules, the average copy number of each of which is about two. Because they do not share the same internal sequences, they could be placed within different families (as defined), but many of them could be the result of capture events mediated by a particular autonomous MULE, which could theoretically mobilize them. Mu elements in maize are a clear example of this. The autonomous element has high homology at the TIRs with the non-autonomous elements but none in the internal sequences. Nevertheless, when the autonomous element is active, all of the non-autonomous elements can replicate. So I’m not sure if they are part of the same “species” or not. Or perhaps the non-autonomous elements can be thought of as parasites? If so, then how does this fit into an ecological model?

Figure 4. What really jumps out for me are two things. First, many of the superfamilies appear to have experience big jumps in copy number recently. Is this really a thing? Second, it really looks like the key distinction with respect to age is at the Family, not the superfamily level. Interesting.

Line 368. A quibble, but saying “partition” suggests that this is a distribution derived from functional constraints rather than a consequence of random loss of function within multiple copies.

line 501. A median of zero is just zero, right?

Line 508. Again, not for this publication, but I wonder if the proportion of converted cytosines in the CG and CHG versus tell a history about a given element. CHH islands tend to be near genes, so a TE lineage that has spent a lot of time near genes could have a different proportion than a TE lineage that didn’t.

Line 550. “...by family, and for example the two..” awkward

Line 562. One wonders if this is because many of the “genes” near helitrons are actually transduplicated parts of the helitron.

567. Again, I’m probably just being thick, but a median of zero means half were above zero and half were below (which they can’t be), so none of them expressed at all?

Figure 6. In legend. D-F?

Line 635. That’s a bit surprising given figure 4, which suggests that there is more variation at the family level.

Line 655. It’s not clear what each line on the Y axis represents here. Correlates with the Y axis names to the left? F-G. what do the colors correspond to? The colors of the labels of families in C? I’m afraid these need a bit more explanation in the legend.

Line 747. And by analysis of de novo, unselected insertions.

Line 779. See also https://bmcgenomics.biomedcentral.com/articles/10.1186/s12864-020-6763-1

Line 794. See my comment earlier about partitioning.

Line 952. “elaborate genome” : )

**Have all data underlying the figures and results presented in the manuscript been provided?**

Reviewer #1: Yes

PLOS authors have the option to publish the peer review history of their article (what does this mean?). If published, this will include your full peer review and any attached files.

Reviewer #1: No

---

## [Editor Report · Decision Letter 2]

10 Aug 2021

Dear Dr Stitzer,

We are pleased to inform you that your manuscript entitled "The genomic ecosystem of transposable elements in maize" has been editorially accepted for publication in PLOS Genetics. Congratulations!

Yours sincerely,

Kirsten Bomblies

Section Editor: Evolution

PLOS Genetics

Kirsten Bomblies

Section Editor: Evolution

PLOS Genetics

Comments from the reviewers (if applicable):

Thanks for your edits! The paper looks really nice and we are happy to see it in our journal.

**Data Deposition**

http://datadryad.org/submit?journalID=pgenetics&manu=PGENETICS-D-19-01176R2

**Press Queries**

---

## [Editor Report · Acceptance letter]

4 Oct 2021

PGENETICS-D-19-01176R2 

The genomic ecosystem of transposable elements in maize 

Dear Dr Stitzer, 

We are pleased to inform you that your manuscript entitled "The genomic ecosystem of transposable elements in maize" has been formally accepted for publication in PLOS Genetics! Your manuscript is now with our production department and you will be notified of the publication date in due course.

With kind regards,

Livia Horvath

PLOS Genetics

On behalf of:
